# The Economic Benefits of Reducing Racial Disparities in Health: The Case of Minnesota

**DOI:** 10.3390/ijerph16050742

**Published:** 2019-03-01

**Authors:** Marilyn S. Nanney, Samuel L. Myers, Man Xu, Kateryna Kent, Thomas Durfee, Michele L. Allen

**Affiliations:** 1Department of Family Medicine and Community Health, University of Minnesota, Minneapolis, MN 55455, USA; msnanney@umn.edu (M.S.N.); miallen@umn.edu (M.L.A.); 2Humphrey School of Public Affairs, University of Minnesota, Minneapolis, MN 55455, USA; xuxx0460@umn.edu; 3Office of Public Engagement, University of Minnesota, St. Paul, MN 55108, USA; kent0082@umn.edu; 4Department of Applied Economics, University of Minnesota, Minneapolis, MN 55455, USA; durfe019@umn.edu

**Keywords:** racial disparities, economic cost, mortality, lost productivity

## Abstract

This paper estimates the benefits of eliminating racial disparities in mortality rates and work weeks lost due to illness. Using data from the American Community Survey (2005–2007) and Minnesota vital statistics (2011–2015), we explore economic methodologies for estimating the costs of health disparities. The data reveal large racial disparities in both mortality and labor market non-participation arising from preventable diseases and illnesses. Estimates show that if racial disparities in preventable deaths were eliminated, the annualized number of lives saved ranges from 475 to 812, which translates into $1.2 billion to $2.9 billion per year in economic savings (in 2017 medical care inflation-adjusted dollars). After eliminating the unexplained racial disparities in labor market participation, an additional 4,217 to 9185 Minnesota residents would have worked each year, which equals $247.43 million to $538.85 million in yearly net benefits to Minnesota.

## 1. Introduction

Across all types of diseases, illnesses, and accidents, Blacks are 1.16 times [1] more likely to die than Whites. Blacks are 1.22 and 1.72 times [2,3,4] more likely than Whites to die from heart disease and hypertension, both preventable diseases. 

According to LaVeist, Gaskin, and Richard [5], the annual cost of racial differences in premature death in the US ranges from $236.1 billion to $243.1 billion. Racial disparities also arise in labor market outcomes. Again, LaVeist, Gaskin, and Richard [5] estimate these costs amount to $11.7 billion to $13.3 billion a year. Other attempts to estimate the economic costs associated with health disparities produce values that range from $193 billion (smoking) to $250 billion (fatal and non-fatal cost of occupation injuries) [6,7,8]. 

Research on the impact of healthcare reform argues for targeted attention to the populations and groups who are at higher risk of incurring high healthcare costs [9,10]. Although there is a case to be made for addressing racial and ethnic disparities in health for population health reasons or social justice [11,12], the novelty of LaVeist, Gaskin, and Richard’ [5] work rests in its ability to make a business case for reducing health disparities in the United States. 

The business case for reducing health disparities might appear more difficult to make in places like Minnesota, which has a relatively low population of racial minority group members. The U.S. Census Bureau’s Population Estimates Program shows that in 2017 non-Hispanic Whites represented 79.9% of Minnesota’s population (https://factfinder.census.gov/faces/tableservices/jsf/pages/productview.xhtml?pid=ACS_17_1YR_CP05&prodType=table). Yet, there are significant racial disparities in major economic and health outcomes in the state. In 2018, U.S. News & World Report ranked Minnesota second in terms of overall best states in which to live in the U.S. At the same time, Minnesota ranked 47th in employment gap by race (ranking best to worst) and 38th in income gap by race. Using the American Community Survey, Myers and Ha [13] point out that Minnesota consistently has had much lower employment rates for racial minorities than elsewhere in the country over the past seventeen years, and also has one of the largest racial disparities in unemployment in the nation. They term this incongruity between Minnesota’s overall high measures of social and economic well-being and large racial disparities in every measure of social and economic well-being as “The Minnesota Paradox.” The same paradox exists in Minnesota’s health outcomes. According to a Commonwealth Fund survey, Minnesota scored second in the nation, second only to Vermont [14], in health system performance. Still, from 2010 to 2014, African Americans were almost two times more likely to have lower birthweight births than Whites; and American Indian infants were more than two times more likely to die than Whites [15,16]. Recognizing the Minnesota paradox, this paper demonstrates that even in locations where there are relatively few racial minorities, there are sizeable economic benefits to be gained from eradicating racial health disparities.

## 2. Materials and Methods 

Estimates of the economic cost of racial disparities are obtained for two health outcomes: mortality rates and labor market effects of illnesses. The analysis of mortality rate disparities conceptually computes the number of lives lost due to different mortality rates between each racial minority group and the lowest mortality rates within each age group and cause of death. In most cases, Whites had the best or second best health profiles, while for age groups 25–34, 35–44, and 45–54, Hispanics/Latinos had the lowest mortality rate. In Minnesota, Hispanics/Latinos comprise approximately 5.3% of the state population (US Census Bureau’s Population Estimates Program, Vintage 2017). To have the most representative estimation, this paper uses the majority racial group, non-Hispanic White, as the reference group. 

This analysis uses the method proposed by LaVeist, Gaskin, and Richard [5] where one converts the mortality rate differentials into number of deaths. The excess deaths due to disparities in mortality rates for each age group and disease equal the real deaths minus the predicted deaths, where the predicted deaths assume each racial and ethnic minority group faces the same risk of death as non-Hispanic Whites within age and type of death categories. This is akin to eliminating all within-age and type of death disproportionalities and also akin to equating the mortality disparity ratio to 1. For some age groups and diseases, some minorities have lower death rates than the total population. We report two calculations: one that retains negative excess deaths, which results in a lower bound of estimates, and another that only keeps the excess deaths, which yields an upper bound of estimates. 

The valuation of these lives lost uses conventional estimates of foregone earnings. The cost of early mortality uses the Value of a Year of a Statistical Life, a proxy for the opportunity cost of treating an underlying condition and an assessment of the benefits of risk reduction efforts [17,18]. This measure is widely used in health research, but with significant variation in the value assigned [19]. This paper uses the most commonly used number $50,000 in 1997 dollars as the lower bound for per quality-adjusted year life and a simulation model based number $61,294 in 2003 dollars as the upper bound for per quality-adjusted year life [19,20]. We consider inflation in the analysis and use the adjusted numbers. The $50,000 in 1997 dollars is $76,316 in 2017 dollars using the Consumer Price Index for All Urban Consumers (CPI-U), and $94,762 using the medical care service subsection of the CPI-U [20]. In addition, $61,294 in 2003 dollars is $81,927 in 2017 dollars using the CPI-U, and $131,387 using the medical care service subsection of the CPI-U). Life expectancy is assumed to be 75 years. To make a per quality-adjusted year life comparable to the estimated value of a statistical life from the United States Environmental Protection Agency (https://www.bloomberg.com/graphics/2017-value-of-life/), we transfer the lower and upper bounds of per quality-adjusted year life into the value of a statistical life for 2017, which results in an estimated range of $5,723,700 to $9,853,999. The upper bound of our adopted value of a statistical life is close to the estimates of the United States Environmental Protection Agency.

The labor market effects of illness are captured by estimating logistic models of the probability of having a work-limiting condition (Equation (1)) [21] and the number of weeks in a year not working given that a person has a work-limiting condition (Equations (3) and (4)). These models control for age, gender, highest level of education achieved, marital status, family structure sectors, and socio-economic covariates, and only include people between 16 and 65 years old. Our labor market participation estimates permit us to estimate the net increases in the number of minorities who would have worked had there been no unexplained disparity in time lost from work (Equation (2)). The log-odds of having a work-limiting condition is given by
(1)$$\ln\left(\frac{\mathrm{Pr}(Y=1|x)}{\mathrm{Pr}(Y=0|x)}\right)=\beta_{0}+\beta_{1}*Minority+\overset{j}{\underset{j=2}{\sum }}\beta_{j}X_{j}{}_{\;}+\varepsilon_{i}\;\;$$
where Y is a dichotomous variable that indicates having a work-limiting condition or not, $X_{j}$ is the set of socio-economic covariates; *Minority* is the dummy variable that represents minority status; and $\varepsilon_{i}$ is the error term. 

Equation (1) is based on the likelihood that a respondent reports having a health condition that limits their ability to work. To identify the degree to which race or ethnicity relate to this measure of health, we estimate the odds of having such a condition for members of a racial/ethnic group ($e^{\beta_{1}})$. Using the estimated coefficient on minority status, we can then compute the additional number of minorities who face limits to their ability to work due to health conditions. This estimate is shown in equation 2 and is equal to the product of the slope of the probability of a work limiting health conduction with respect to minority status and the number of minorities. Denote $\Delta \left(N_{m}\right)$ as the additional number of minorities faced with work limitations due to health conditions. Then,
(2)$$\Delta \left(N_{m}\right)=\hat{\beta_{1}}*p*\left(1-p\right)*N_{m}$$
where p is the probability of having a work-limiting condition, $\hat{\beta_{1}}$ is the estimated coefficient on minority status from the logistic model, and $N_{m}$ is the estimate of the minority working-age population in Minnesota.

Equation (2) estimates the number of minorities who would have worked had there been no racial disparity in work-limitations due to health, controlling for other determinants of work-limiting conditions. First, we estimate the change in an individual’s probability of having a work-limiting condition according to minority status. This change is obtained from the estimated coefficients in the logistic model where the derivative of the probability of a work-limited condition with respect to the minority status is $\frac{\partial p}{\partial \mathrm{Minority}}=\beta_{1}*\left(1-p\right)*p$. Second, we multiply this estimate by the size of the estimated minority working-age population (working age range from 16 to 65). The result is an estimate of the population-wide economic effect of reducing health disparities in labor market participation. After estimating the number of minorities that would be affected by the possibility of equal health treatment, we perform a Blinder–Oaxaca decomposition [22,23,24] to obtain equal treatment estimates for the number of weeks worked in a year, according to minority status and according to our controls (Equation (3)). 

The Blinder–Oaxaca decomposition proceeds by estimating separately for minorities and non-minorities (Equations (3) and (4)) the number of missed weeks from work due to a work limiting health condition. We then estimate the number of missed weeks from work for minorities when they are treated like non-minorities (Equation (5)) by applying the coefficients from Equation (4) to the independent variables in Equation (3). 

The number of missed work weeks for minorities is given by:(3)$$Z_{Y=1}^{m}=\beta^{m}+\sum_{i=1}^{k}\beta^{m}{}_{i}\cdot X^{m}{}_{i}+\varepsilon_{i}$$

The number of missed work weeks for non-Hispanic whites is given by:(4)$$Z_{Y=1}^{n}=\beta^{n}+\sum_{i=1}^{k}\beta^{n}{}_{i}\cdot X^{n}{}_{i}+\varepsilon_{i}$$

The number of missed work weeks with equal treatment of minorities and non-minorities is given by:(5)$${\tilde{Z}}_{Y=1}^{m}={\hat{\beta }}^{n}+\sum_{i=1}^{k}{\hat{\beta }}_{i}^{n}\cdot {\bar{X}}_{i}^{m}+\varepsilon_{i}$$
where Y is a dichotomous variable that indicates having a work-limiting condition; Z is the number of missed work weeks; and the superscripts m and n denote minorities vs. non-minorities. We then multiply this aggregate number of missed work weeks per year by the mean earnings for a typical Minnesotan. The result is a base estimate of the value of the missed work weeks per year that minorities face that cannot be explained by differences in the characteristics of minorities vs non-minorities.

When someone is absent because of an illness the employer faces an unfilled job slot. Employers must fill this temporary vacancy by either finding a replacement worker, paying their remaining workers overtime, or by scaling back their production. To measure the firm’s cost of filing a vacant position, we rely on the literature on the economics of job searches.

Firms are willing to incur costs to fill vacant positions to avoid scaling back their production [25]. These costs include both the accounting costs of posting their vacant position and the opportunity costs the firm faces by reducing their production until they find a suitable replacement.

Russo et al. [25] find that on average, the cost of filling a vacancy is approximately 3% of the cost of the earnings of the position itself. We use their baseline estimate of 3% for this study, to capture the average cost of vacancy throughout a workforce. To account for this 3% search cost of vacant positions faced by employers, we scale our labor cost estimates by 1.03. 

### Data

The estimates provided herein utilize two publicly available databases: Integrated Public Use Microdata Series-USA(IPUMS-USA) [26] and the Minnesota Center for Health Statistics Vital Records on Mortality (years 2011–2015). Among the demographic variables used in the models are age, military service history, gender, race, ethnicity, head of the household, and highest level of education achieved. Among the economic variables are individual income, household income, federal household poverty status, number of weeks worked in the last year, the presence of a work-limiting condition which indicates whether respondents have any lasting physical or mental health condition that causes difficulty working, limits the amount or type of work they can do, or prevents them from working altogether and public sector employment. The sample was limited to Minnesotans of working age (over 15 and under 65).

The Minnesota Center for Health Statistics provided administrative data that represent the population of Minnesota residents with death certificates in the years 2011–2015. Cause of death is reported according to the International Classification of Disease (ICD-10), race/ethnicity, and national origin, age, gender, marital status, level of education, and geography in the form of a zip code. Only Minnesota residents for whom a death certificate is available were included in the analysis. If a Minnesota resident died in another state, the circumstances of their death are still reflected in the data. However, information about non-Minnesota residents who died in Minnesota is not included in the data. We aggregate these populations to form estimates of the age-adjusted mortality rate by a given cause of death.

## 3. Results

Between 2011 and 2015, there were 204,723 deaths with certification numbers recorded in Minnesota (Center for Health Statistics Vital Records on Mortality Data, 2011–2015). Of that total, 12,306 or about 6% were listed as American Indian, African American, or Asian American, and 2051 or 1% were listed as Hispanic/Latino. Of these 12,306, 5819 (2.84%) were African Americans; 2773 (1.35%) were American Indians; 2951 (1.44%) were Asian Americans or Pacific Islanders; and 763 (0.37%) were Other races. Table 1 shows that the three ethnic groups with the highest overall mortality rates are American Indian, African American (not African), and Southeast Asian (Vietnamese, Bhutanese, Cambodian, Hmong, Laotian, Thai, and Burmese) or unspecified Asian. In comparison to the age-adjusted mortality rates for Whites, age-adjusted mortality rates are twice as high for American Indians; 1.19 times as high for African Americans; and 1.02 times as high for Southeast Asians.

Asians and Pacific Islanders have lower overall mortality rates than White Non-Hispanics, which means there should not be any excess deaths from these two race/ethnic groups. However, if we focus on particular age groups, Asian and Pacific Islander children under 15 years old have higher mortality rates than their White Non-Hispanic peers. As shown in Figure 1, the disparity ratio between Asian and Pacific Islander children to all Minnesota children from 5 to 14 years old is 4.94. American Indians in Minnesota experience the worst health disparities of all ethnic/racial groups. They have the highest mortality disparity ratios for chronic liver diseases, influenza and pneumonia, diabetes, motor vehicle and other accidents, and homicide (Appendix Table A1). Sarche and Spicer [27] linked these adverse health outcomes to poverty and barriers to employment due to geographic isolation and lack of employment opportunities. Meanwhile, non-Hispanic Whites experience higher mortality rates than Hispanics/Latinos in Minnesota, with a 711.79 per year mortality rate for the non-Hispanic White population and 527.21 for Hispanics. However, the Hispanic/Latino population experiences mortality disparities in chronic liver disease and homicide (Appendix Table A1). 

As Table 2 shows, a large number of lives could be saved among minorities depending on the model used, indicating that the differences between minority groups play an important role in defining these estimates. If we break down potential lives saved by reducing illness-related and non-illness-related causes (including homicide, suicide, motor vehicle accidents, other accidents, and other external causes), then 397 to 629 minority lives could be saved per year, while 78 to 183 minority lives could be saved by reducing non-illness-related causes. The associated per year economic benefits for lives saved among minorities is presented in Figure 2 and the range of the resulting estimates can be found in Appendix Table A2. Once the health disparities for minorities are eliminated, the lower bound of the value that can be saved is $1.226 billion and the upper bound of the value that can be saved is $2.940 billion.

From 2005 to 2007, in Minnesota, 164,396 persons did not work at any point during a year as a result of illness. Of those, 29,103 or 17.7% were minorities. A non-Hispanic White worker with an illness missed an average of 43.46 weeks whereas minorities missed 46.13 weeks on average. This suggests that although there are some who may continue to work while they have a work limiting condition, most miss work altogether. As seen in Table 3, these differences in the number of weeks lost due to illness resulted in African Americans, American Indians, Asians and Pacific Islanders, as well as persons of other or multiple races missing more work than their White counterparts. 

Table 4 presents our logistic model estimation of the probability of missed work due to illnesses obtained a range of 4,217 to 9,185 additional minorities who would have worked had there been no racial disparity. Appendix Table A3 illustrates the underlying logistic models, in which controls include age, gender, head of household, household poverty, education, and military service. In short, 0.91% to 1.99% minorities of working age would have worked each year had there not been a racial disparity. For those additional minorities who would have worked, the average number of weeks persons would have worked in 2007 is 39.98 weeks, which equals the real average number of weeks worked in the year of 2007 for Minnesotans. For the state of Minnesota, the average estimated economic benefits of increased work had there been no unexplained racial disparities in weeks not worked due to illness is $427.33 million. 

Table 5 illustrates the Blinder–Oaxaca decomposition analysis of the number of extra weeks that minorities would have worked had there been no unexplained racial disparity. The estimates yield between 1.72 and 2.12 extra work weeks, which equals 4.3% to 5.3% of the average number of work weeks for Minnesotans. The covariates included in the decomposition analysis are age, highest achieved educational level, gender, household poverty, military service history, household head, and employment in the public sector. 

## 4. Discussion

This paper examines health disparities that affect minority groups in Minnesota and estimates the economic benefits if these disparities were eliminated. The findings establish that most racial/ethnic groups in Minnesota experience health disparities. Should those disparities be eliminated, the state economy could see at least an additional $1.226 billion in economic activity. The savings from work-limiting health conditions that impact the number of weeks minorities miss during a year could bring over $247.43 million to the local economy. Economic gain aside, the problem of health disparities must be addressed because it affects the quality of individual lives and entire communities in Minnesota and across the nation. Progress has been made at the state and national levels targeting and improving some health disparities, such as infant mortality rates [28,29], but many still persist. This study addresses the economic costs of health-related disparities and does not address well-documented structural barriers [30,31] and other determinants of health. Therefore, the cost estimates are conservative.

Meanwhile, local media and advocates in Minnesota foresee and have concerns about tomorrow’s workforce shortages [32]. “Minnesota’s aging workforce has tightened the current job market near its ‘full potential,’ meaning nearly one job for every applicant. Finding enough high-skilled workers will be among the top issues that face Minnesota’s biggest companies in coming years.” Once one connects the current excess deaths among minorities and the expected future labor force shortages, the estimated economic costs for racial health disparities are even more conservative.

The estimates from this paper reflect a set of social determinants of health, associated with labor market participation and time lost due to health conditions. However, on the basis of the available data, we cannot perform an analysis for the disaggregated illnesses. Additionally, there may be factors that contribute to health that are correlated with race, ethnicity and national origin that are not included in the list of variables used in the analysis. For instance, it is well-documented that historical trauma and chronic stress are significant factors that influence the health of African American and Native American communities [33,34]. However, it is not possible to measure these variables with existing databases.

The decomposition analysis focuses on race and ethnicity. We control for gender in the regressions but we do not decompose the analysis separately by gender within racial and ethnic groups. Decomposition by gender might produce different estimates of the overall costs to society of racial and ethnic disparities in health.

Another important variable not included in the analysis is immigrant status. For instance, interaction between health and employment of recent immigrants from African may be different than that for the overall African American or African population of Minnesota. 

From the overall mortality rate, we see that non-Hispanic Whites have higher mortality rates than Hispanics/Latinos in Minnesota. This finding is consistent with research on the Hispanic/Latino population nationwide and is known as the Hispanic or Latino Paradox [35,36]. The Hispanic/Latino Paradox states that Hispanics/Latinos have mortality and morbidity advantages over White Non-Hispanics due to positive selection via immigration [37]. The paradox arises because even though Hispanics/Latinos as a group have poorer social and economic outcomes newer arrivals are in better health. However, the research agrees that US-born Hispanics/Latinos experience a higher rate of mortality from certain cancers and chronic liver disease than their foreign-born counterparts [38]. Resilience, diet and nutrition, and social support and cohesion are known to be possible assets contributing to lower mortality rates among foreign-born Hispanics/Latinos [39]. 

In short, one important limitation of the analysis is our inability to flesh out empirically the possible immigration effects and interactions with race and ethnicity. As in other research where testing the Healthy Migrant Effect (HME) has proven to be difficult, this paper does not disentangle the interactions between ethnicity, race, and migration [40,41,42,43]. If anything, the HME might bias downward our measures of cost of lives saved and weeks worked through reductions in health disparities. Further research might uncover how immigration status affects estimates of number of lives saved and the economic benefits derived through reductions in racial/ethnic disparities in health.

## 5. Conclusions

Our results add to the health equity literature and provide policymakers with another tool to address persistent health inequities: even in Minnesota, with a relatively small minority population, there are substantial costs associated with racial health disparities. The costs accrue through lives lost and fewer weeks worked due to illness. The nontrivial cost savings via reductions in racial health disparities suggest that public and private investments can be justified to produce targeted improvements in the health of racial and ethnic minority group members. 

## Figures and Tables

**Figure 1 ijerph-16-00742-f001:**
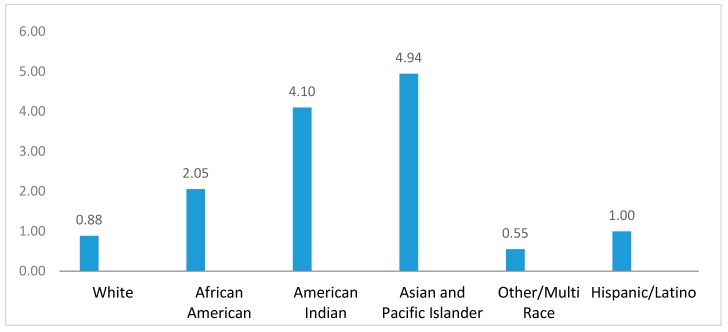
Disparity ratios for mortality rate, children ages 5–14.

**Figure 2 ijerph-16-00742-f002:**
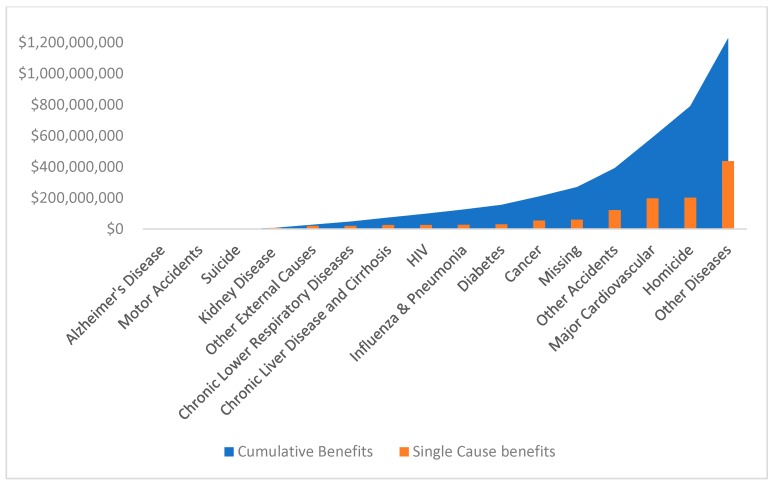
Annualized economic benefits for saved lives (2011–2015), lower bound.

**Table 1 ijerph-16-00742-t001:** Age-adjusted overall mortality rate by race/ethnicity, in Minnesota (MN), 2011–2015.

Race/Ethnicity	2011–2015 Overall Mortality Rate in MN per 100,000 persons	Disparity Ratios for Overall Mortality Rate
White	711.79	1.00
African American, all	849.94	1.19
African American, NOT African	848.15	1.19
Western African	317.35	0.45
Eastern African	404.40	0.57
American Indian	1589.34	2.23
Asian and Pacific Islander	573.33	0.81
Southeast Asian or Unspecified Asian	727.44	1.02
Asian and Pacific Islander, NOT Southeast Asian or Unspecified Asian	365.88	0.51
Other/Multi race	269.07	0.38
Hispanic/Latino (any race)	527.21	0.74

**Table 2 ijerph-16-00742-t002:** Annualized potential lives saved among minorities by cause of death.

Cause of Death	African American	American Indian	Asian and Pacific Islander	Other Races	Hispanic/Latino (Any Race)
Alzheimer’s Disease	0 to 0	0 to 0	−1 to 0	−1 to 0	−1 to 0
Cancer	60 to 68	37 to 43	−20 to 5	−41 to 0	−12 to 1
Chronic Liver Disease and Cirrhosis	0 to 3	22 to 23	−8 to 0	−4 to 0	−13 to 3
Chronic Lower Respiratory Diseases	11 to 12	12 to 13	−8 to 1	−7 to 0	−7 to 3
Diabetes	13 to 14	13 to 13	−1 to 2	−3 to 0	−7 to 2
HIV	10 to 10	1 to 1	0 to 0	0 to 0	0 to 2
Homicide	43 to 43	8 to 8	2 to 2	1 to 2	1 to 7
Influenza and Pneumonia	5 to 5	7 to 7	−1 to 1	−2 to 0	−1 to 7
Major Cardiovascular	91 to 95	56 to 60	0 to 6	−23 to 0	−5 to 1
Missing	22 to 22	13 to 14	7 to 7	−1 to 1	−5 to 5
Motor Accidents	−3 to 1	12 to 13	−5 to 1	−9 to 0	0 to 5
Kidney Diseases	7 to 7	2 to 2	0 to 1	−1 to 0	0 to 2
Other Accidents	39 to 41	44 to 45	−18 to 2	−18 to 0	−5 to 1
Other Diseases	103 to 109	65 to 69	2 to 15	−40 to 0	−6 to 5
Other External Causes	4 to 5	4 to 4	−1 to 1	0 to 0	−4 to 5
Suicide	−8 to 2	10 to 12	−9 to 1	−18 to 0	−17 to 1

**Table 3 ijerph-16-00742-t003:** Weeks of work missed due to illness, ages 16–65 (2005–2007).

Race/Ethnicity	Missed Weeks	Disparity Ratio	SD	Weighted *N*
White	43.43	0.9887	15.90	137,578
African American	46.99	1.0696	12.07	11,703
American Indian	48.02	1.0930	10.42	4127
Asian and Pacific Islander	45.99	1.0468	14.38	5676
Other/Multi Race	44.71	1.0177	13.34	5312
Hispanic/Latino (Any Race)	40.56	0.9232	15.94	4838
**Binary Classification**				
White, Non-Hispanic	43.46	0.9893	15.91	135,293
Minority	46.13	1.0500	12.92	29,103
Total	43.93	1.0000	15.45	164,396

**Table 4 ijerph-16-00742-t004:** Annualized additional number of minorities who would have worked had there been no disparity, ages 16–65 (2005–2007).

Model Specification	$\beta_{minority}$	1−p	p	Weighted MN Minority Working Age People $\left(N_{m}\right)$	Additional number of minorities who would have worked $\left(\Delta N_{m}\right)$
**Model 1**	0.3798	0.9446	0.0554	462,109	9185
**Model 2**	0.3494	0.9446	0.0554	462,109	8449
**Model 3**	0.1744	0.9446	0.0554	462,109	4217
**Average**					7284

**Table 5 ijerph-16-00742-t005:** Blinder–Oaxaca decomposition of difference in weeks missed due to illness, between Non-Hispanic Whites and minorities, ages 16–65 (2005–2007).

Predicted Missed Weeks of Work	Model 1	Model 2	Model 3	Model 4
	b/(t)/[se]	b/(t)/[se]	b/(t)/[se]	b/(t)/[se]
Minority	46.1282 ***	46.1282 ***	46.1282 ***	46.1282 ***
	(70.4467)	(70.3669)	(70.7899)	(70.8121)
	[0.6548]	[0.6555]	[0.6516]	[0.6514]
White, non-Hispanic	43.4605 ***	43.4605 ***	43.4605 ***	43.4605 ***
	(128.0687)	(129.3139)	(128.1400)	(129.4115)
	[0.3394]	[0.3361]	[0.3392]	[0.3358]
Difference	2.6678 ***	2.6678 ***	2.6678 ***	2.6678 ***
	(3.6173)	(3.6214)	(3.6316)	(3.6401)
	[0.7375]	[0.7367]	[0.7346]	[0.7329]
Explained Portion	0.9507 **	0.7317	0.6510 **	0.5507 *
**(%)**	(35.64%)	(27.43%)	(24.40%)	(20.64%)
	(2.2006)	(1.6247)	(2.3007)	(1.7758)
	[0.4320]	[0.4503]	[0.2829]	[0.3101]
Unexplained Portion	1.7171 *	1.9361 **	2.0168 ***	2.1170 ***
**(%)**	(64.36%)	(72.57%)	(75.60%)	(79.35%)
	(1.9108)	(2.2135)	(2.7316)	(2.9133)
	[0.8986]	[0.8747]	[0.7383]	[0.7267]
Minorities (*N*)	556	556	556	556
White, non-Hispanics (*N*)	4306	4306	4306	4306

Robust Standard Errors in Parentheses. *** *p* < 0.01, ** *p* < 0.05, * *p* < 0.1.

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
