# Peer review of "The Economic Benefits of Reducing Racial Disparities in Health: The Case of Minnesota"

_ijerph, 2019, doi:10.3390/ijerph16050742_

Reviewer 1 Report

Given the topic, I am disappointed that the authors did not include analysis by gender. 

Author Response

Analysis by Gender:

We agree with the reviewer that it would have been ideal to disaggregate the analysis by gender.  However, we consider this to be a larger undertaking requiring a completely different analysis and methodology since gender differences in labor markets would require re-estimation of all of the equations and partitioning the results accordingly. Given the request that we return the paper in 7 days, we were unable to perform these revisions. We note, however, in the limitations section that accounting explicitly for gender differences might reveal different results.

We do note, however, that we control for gender in the underlying regressions.

Reviewer 2 Report

This paper addresses an important topic and is well-written. I have some concerns for the authors to consider in revising the manuscript. 

The methods the authors used does not seem to account for some of the important confounding factors such as immigration status. For instance, the difference between mortality rates of Hispanic and White may be an artifact of healthy immigrants hypothesis. Who are included in the definition of African Americans -- I assume that it might include recent immigrants such as the Somali communities. The interaction between health and employment among Somali communities may be different from that of the other African Americans. 

The authors did not seem to factor in the adjustment of labor market -- if one exits labor market due to death or morbidity, there might be another person who fill in the position. Of course, this relates to the debate between human capital approach and the frictional cost approach in calculating lost productivity but this might be particularly relevant to this paper.

The authors calculated a value of statistical life by using the $50,000 threshold of QALY -- why? There are current estimates that you can use directly -- and they are very different from what the authors calculated. Examples can be found at https://www.bloomberg.com/graphics/2017-value-of-life/. 

Author Response

1.      The methods the authors used does not seem to account for some of the important confounding factors such as immigration status. For instance, the difference between mortality rates of Hispanic and White may be an artifact of healthy immigrants hypothesis. Who are included in the definition of African Americans -- I assume that it might include recent immigrants such as the Somali communities. The interaction between health and employment among Somali communities may be different from that of the other African Americans.

In our discussion section, we acknowledge the possible bias in our estimates of the lives saved and weeks worked by not explicitly accounting for differential impacts on immigrants vs. non-immigrants. We note some of the literature that points to the difficulties in parsing out the Healthy Migrant Effect (HME) put speculate that our estimates might be biased downwards as a result.  We appreciate the Reviewer’s expression of caution in interpreting the results.

2.      The authors did not seem to factor in the adjustment of labor market -- if one exits labor market due to death or morbidity, there might be another person who fill in the position. Of course, this relates to the debate between human capital approach and the frictional cost approach in calculating lost productivity but this might be particularly relevant to this paper.

In the revision, we account for the labor market effects of finding replacement labor in the case of lost labor. We address this comment by appealing to the literature on Search Frictions in the labor market faced by the firm in order to fill a vacant position. Although this literature does not distinguish between vacancies created due to mortality and other causes of vacancies, we find in the literature a sufficient base line estimate to address this concern. The costs measured in the revisions include accounting costs and opportunity costs. We use an adjustment factor of 1.03.

3.      The authors calculated a value of statistical life by using the $50,000 threshold of QALY -- why? There are current estimates that you can use directly -- and they are very different from what the authors calculated.

We have recalculated the value of a statistical life using alternative measures cited in the literature.  The most commonly used measure is the $50,000 value adjusted for inflation and represents a lower bound. Other measures calculated in the revisions include the EPA methodology adopted from Lee et al. (2009) [1] The revised results report the upper and lower bounds.

[1]  https://www.bloomberg.com/graphics/2017-value-of-life/ and Lee, C. P.; Chertow, G. M.; Zenios, S. A. An empiric estimate of the value of life: Updating the Renal Dialysis Cost‐Effectiveness Standard. Value Health 2009, 12(1), 80-87.

Reviewer 3 Report

#1. The reviewer can’t really understand equation-2 and table 4.

  By table 4, Nminority = 101,434, so [take model-1 as an example] “Additional Number of minorities” = beta*(1-p)* Nminority = 0.3798*0.9446* 101434 [by Equation -2] = 36390, much larger than 2925 as reported in table 4.

  Actually in the reviewer’s mind, the Equation 2 should be beta*p*(1-p)* Nminority [because the “p” of [minority/p] in footnote 5 was unnecessary]. If that is the case, then “Additional Number of minorities” would be 0.3798*0.0554*0.9446* 101434= 2016,  lower but closer to 2925 as reported in table 4.

#2. ref-21 21. Minnesota Department of Health. Populations of Color Health Update: Birth and Death Statistics. 315 Available online: https://seedsofnativehealth.org/wp-content/uploads/2017/05/Populations-of-Color-316 Health-Update-Birth-and-Death-Statistics.pdf (accessed on December 2015). 317 >> “Page Not Found” when accessed Jan 5th 2019

Author Response

1.      The reviewer can’t really understand equation-2 and table 4.

We have corrected and rewritten equation 2 and provided a more detailed explanation. We have also edited all of the other equations to make them more readable. We have updated the results in Table 4.

2.      For Reference 21 (Minnesota Department of Health. Populations of Color Health Update: Birth and Death Statistics.) the page was not found

We have updated the link for this reference.

Round  2

Reviewer 2 Report

The authors have addressed my concerns and I have no further comments.

Author Response

Thank you for accepting our revisions based on your comments.

Reviewer 3 Report

#1 line 20 -21 “the annualized number of lives saved ranges from 475 to 812” & line 202 -203 “illness-related causes, then 397 to 629 minority lives could be saved per year, while 94 to 183 minority lives could be saved by reducing non-illness-related causes” >> 812 = 629 + 183, but 475 < 491 = 397+94, please clarify

#2 line 21 “475 to 812, which translates into $1.2 billion to $2.9 billion per year in economic savings” & line 86 -87 “the value of statistical life 2017, the estimated range is from $5,723,700 to $9,853,999” >> Was the value of statistical life age-dependent? If not, then 475*5723700 = roughly 2.7 billion, much higher than 1.2 billion. Please clarify

#3 line 205 -206 “the lower bound of value that can be saved is $1.226 billion and the lower bound of value that can be saved is $2.940 billion” >> or, “the lower bound of value that can be saved is $1.226 billion and the upper bound of value that can be saved is $2.940 billion.”?

#4 table 4 “Weighted MN Minority Working Age People (Nm)”: In the last version, it should be 8662 [=101434*0.0854] whereas 462109 was reported in the current version. Please clarify [due to typo-error in the previous version?]

Author Response

REVIEWER #3

#1 line 20 -21 “the annualized number of lives saved ranges from 475 to 812” & line 202 -203 “illness-related causes, then 397 to 629 minority lives could be saved per year, while 94 to 183 minority lives could be saved by reducing non-illness-related causes”

             >> 812 = 629 + 183, but 475 < 491 = 397+94, please clarify.

We thank the reviewer for pointing out this error. The numbers 475 to 812 in lines 20-21 are correct. However, instead of 94, the lower bound of non-illness related causes should be 78. We apologize for this error and have updated the number in line 202.

#2 line 21 “475 to 812, which translates into $1.2 billion to $2.9 billion per year in economic savings” & line 86 -87 “the value of statistical life 2017, the estimated range is from $5,723,700 to $9,853,999”

            >> Was the value of statistical life age-dependent? If not, then 475*5723700 = roughly 2.7 billion, much higher than 1.2 billion. Please clarify.

The value of the statistical life is the same across ages; however, people died at different ages. Therefore, the foregone benefit is different for different ages. The economic benefit is much higher for a child who died at the age of 5 than for a senior person who died at the age of 70. When we calculate the economic savings, we calculate the yearly savings between the age of death and the age of premature death (75 years old). Appendix Table A2 presents the estimated economic benefits at different ages of death.

#3 line 205 -206 “the lower bound of value that can be saved is $1.226 billion and the lower bound of value that can be saved is $2.940 billion”

            >> or, “the lower bound of value that can be saved is $1.226 billion, and the upper bound of value that can be saved is $2.940 billion.”?

We apologize for this typo. The correct version is “the lower bound of the value that can be saved is $1.226 billion, and the upper bound of the  value that can be saved is $2.940 billion”. We have updated the number in line 206.

#4 table 4 “Weighted MN Minority Working-Age People (Nm)”: In the last version, it should be 8662 [=101434*0.0854] whereas 462109 was reported in the current version. Please clarify [due to typo-error in the previous version?]

In the previous version, 101,434 was the American Community Survey sample size for Minnesota Working-Age People, and 0.0854 was the Minority share of the sample size. After the first round of feedback from all three reviewers, we updated the analysis to reflect the weighted working age of people in Minnesota. Therefore, 462,109 is the weighted Minnesota minority working people number from the American Community Survey. We thank the reviewer for pointing out the inconsistency.